# Elucidation of the Roles of Ionic Liquid in CO_2_ Electrochemical Reduction to Value-Added Chemicals and Fuels

**DOI:** 10.3390/molecules26226962

**Published:** 2021-11-18

**Authors:** Sulafa Abdalmageed Saadaldeen Mohammed, Wan Zaireen Nisa Yahya, Mohamad Azmi Bustam, Md Golam Kibria

**Affiliations:** 1Chemical Engineering Department, Universiti Teknologi PETRONAS, Seri Iskandar 32610, Perak, Malaysia; sulafa_19001261@utp.edu.my (S.A.S.M.); azmibustam@utp.edu.my (M.A.B.); 2Centre for Research in Ionic Liquid, Universiti Teknologi PETRONAS, Seri Iskandar 32610, Perak, Malaysia; 3Department of Chemical and Petroleum Engineering, University of Calgary, 2500 University Drive, NW, Calgary, AB T2N 1N4, Canada; md.kibria@ucalgary.ca

**Keywords:** ionic liquids, CO_2_ electrochemical reduction, electrolyte, co-catalyst

## Abstract

The electrochemical reduction of carbon dioxide (CO_2_ER) is amongst one the most promising technologies to reduce greenhouse gas emissions since carbon dioxide (CO_2_) can be converted to value-added products. Moreover, the possibility of using a renewable source of energy makes this process environmentally compelling. CO_2_ER in ionic liquids (ILs) has recently attracted attention due to its unique properties in reducing overpotential and raising faradaic efficiency. The current literature on CO_2_ER mainly reports on the effect of structures, physical and chemical interactions, acidity, and the electrode–electrolyte interface region on the reaction mechanism. However, in this work, new insights are presented for the CO_2_ER reaction mechanism that are based on the molecular interactions of the ILs and their physicochemical properties. This new insight will open possibilities for the utilization of new types of ionic liquids. Additionally, the roles of anions, cations, and the electrodes in the CO_2_ER reactions are also reviewed.

## 1. Introduction

Since industrialization, fossil fuels have been chiefly used as the primary source of energy, which has led to global environmental problems due to the carbon dioxide (CO_2_) emissions to the atmosphere. Thus, the mitigation of CO_2_ emissions has become a serious issue globally [1,2]. Carbon capture, storage, and utilization (CCSU) is a pivotal way for reducing emissions in industries. Several CO_2_-utilization technologies using biological and chemical methods such as thermal reduction [3], electrochemical reduction [4], and photoelectrochemical reduction [5] have been investigated. The thermal reduction of CO_2_ thermal reduction is far from use in real-world applications because of the high threshold requirements for the temperature, the available materials, and the system configurations [6]. Many techniques have been studied methods to reduce the threshold temperature and to improve the conversion efficiency of the CO_2_ reduction reaction, such as the through the utilization of heterogeneous catalysts [7]. The development of heterogenous catalysts have focused on modifying the structure and the composition, the study of reaction pathways considering dimension–geometry effects, bifunctional processes, ligand impacts, and lattice strain [6]. Several metal catalysts have recently been designed that have shown CO_2_ methanation at low temperatures and at low atmospheric pressure [8]. However, the thermal reduction of CO_2_ thermal is still a big challenge.

The major interest in adopting the electrochemical reduction of CO_2_ is its potential integration with renewable energy such as wind and solar energy, as shown in Figure 1. Moreover, it can be operated under ambient conditions, and the reactions can be easily controlled by adjusting external parameters such as the electrolytes, the type of electrodes, and the applied voltages. Moreover, several studies have reported the use of solar energy for CO_2_ electrochemical reduction (CO_2_ER) both directly and indirectly through photocatalytic chemistry [9,10,11,12], photo-electrochemical [13,14,15], and electrochemical systems [16,17].

Different configurations have been used as a reactor for the CO_2_ electrochemical reduction reaction, which have been inspired by water electrolyzers (liquid phase, solid oxide, and gas phase) [18]. In each type of reactor, the CO_2_ electrochemical reduction reaction (CO_2_ERR) occurs on the cathode side, while the water oxidation reaction takes place on the anode side. In liquid-based electrolytes, the typical CO_2_ reduction cells are traditional H-cells, as depicted in Figure 2, and flow cells, as illustrated in Figure 3 [19]. In the H-cell configuration, the cell consists of an immersed anode and cathode in an electrolyte that have been separated from each other by an ion-exchange membrane. The membrane only allows hydrogen ions to flow into the cathode side, where it prevents the product that is produced in the cathode side from flowing to the anode side and from being oxidized again. Furthermore, the ion-exchange membranes prevent the evolved O_2_ in the anode side from passing across the cathode and consuming the electrons for an oxygen reduction reaction (ORR) that could otherwise be utilized for CO_2_ERR. On the anode side, the water oxidation reaction occurs, producing the hydrogen ions and electrons that will be transferred to the cathode side where the CO_2_ reduction reaction takes place. In the flow cell (Figure 3), the liquid electrolyte is in a flow-through configuration to increase CO_2_ solubility, minimize mass transport limitations, and inhibit hydrogen evolution reactions (HER) [19,20].

Other design configurations are shown in Figure 4 [18]. The liquid-phase electrolyzer has three flow streams: CO_2_, catholyte, and anolyte each have their own stream. The catholyte and gas channel are separated by a gas diffusion electrode. The electrolyte is in contact with a catalyst layer, which is located on the liquid-facing front part of the gas diffusion electrode. In contrast, from the back of the electrode, the CO_2_ gas stream is continuously supplied. This configuration enables the accurate control and enhancement of the reaction environment. A liquid electrolyte system can be advantageous; however, it can also be a source of system instability due to contaminant accumulation on the catalyst and the potential permeation of the contaminant into the gas diffusion electrode; it may also cause flooding, which is a widespread type of failure.

In gas-phase reactors, a solid polymer electrolyte separates the cathode and anode (ion-exchange membrane). In a zero-gap configuration, the cathode catalyst is pressed up against the ion-exchange membrane. The humidity must be supplied to the system via a liquid electrolyte on the anode and/or humidification of the gas inlet stream to ensure that the membrane is hydrated while it is in use. The advantage of gas-phase electrolyzers over liquid-phase configurations is that they have less ohmic losses, less electrolyte pumps and flow fields are used, and they can easily be pressurized. In addition, the absence of the catholyte eliminates multiple sources of instability, such as the accumulation of electrolyte impurities on catalysts and the electrolyte inundating the gas diffusion electrode. On the other hand, liquid products can accumulate in the gas diffusion electrodes, obstructing gas diffusion [18].

The solid oxide flow cell consists of a solid cathode, anode, and electrolyte. Lanthanum gallate-based oxides, zirconia-based oxides, or ceria-based oxides are used as solid electrolytes. This configuration requires a high temperature (above 600 °C) in order to produce a C_1_ gas product (CO/CH_4_). This setup is highly stable, and it runs at la ow voltage and at high current densities [18]. However, the product’s limited range and the excessive temperature demand limits it from being widely used [21,22,23,24].

The processing of carbon dioxide into more complex molecules through chemical catalysis is conducted on a grand scale by photosynthesis using enzymes and radiation. This effective natural system has stimulated a massive amount of work to develop catalysts to perform as enzyme mimics [25,26,27,28,29,30,31,32]. The electrochemical reduction of carbon dioxide (CO_2_) to produce various organic compounds such as methane (CH_4_) [33], methanol (CH_3_OH) [34,35], carbon monoxide (CO) [36], ethylene(C_2_H_4_) [37], and formic acid (HCOOH) [38] have been reported. During CO_2_ reduction, different products can be obtained on the cathode side, which is based on the number of electrons that is required at the respective reduction potentials shown in Table 1. Moreover, the hydrogen ions that have been dissolved in the catholyte might also be converted to hydrogen (H_2_), causing a hydrogen evolution reaction (HER), which is a competitive reaction to CO_2_ERR. This also affects the selectivity, which primarily depends on the number of electrons and catalytic active sites as well as the reactants/intermediates adsorption/desorption properties [39]. Thus, the practical potentials needed for CO_2_ERR are more negative than the equilibrium potentials. Thermodynamically, a high-energy barrier is required for the initiation of the CO_2_ radical (CO_2_^•−^) with an equilibrium potential of −1.90 V vs. a standard hydrogen electrode (SHE), pH = 7), which influences the CO_2_ERR significantly [40].

In CO_2_ERR, the faradaic efficiency must be considered since it represents the efficiency with which charges (electrons) are transported in a system, promoting an electrochemical reaction. On the other hand, the current density, which is defined as the rate of charge passed across a specified cross-section unit area per unit time [42], is affected by the conductivity and thickness of the thin film on the electrode [43]. The main problem associated with this technology is finding a catalytic system that can efficiently convert CO_2_ to the desired value-added product at a low energy requirement. This is due to the linear structure of CO_2_, which makes it electrochemically stable. This results in a process with low overpotential, high faradaic efficiency, high current density, high selectivity towards a specific product with a good yield, and high CO_2_ solubility in the electrolyte.

Many reviews have reported on the role of the electrode as a catalyst employing metals such as copper [44], silver [45], gold [46], lead, and indium [47], alloys at different scales (nano, micro), structures, and morphology [48,49,50], and coated metals [51] as well as metal oxides [52]. Hori et al. reported the effects of different metals on CO_2_ERR and highlighted the electrode’s role in terms of selectivity and reducing the overpotential [53]. The difficulties that are faced during the adsorption of CO_2_ on the electrode surface are the reason for the increased amount of energy that is required for the activation of the reaction [39,54]. On the other hand, in terms of the electrolytes, potassium bicarbonate salt has been widely employed [55,56]. Dunwell and Lu [57] reported that bicarbonate salt has a significant function in the CO_2_ERR instead of merely acting as a pH buffer or proton donor. They suggested that the bicarbonate raises the CO_2_ERR rates by increasing the effective reducible CO_2_ concentration in a solution. A growing interest in ionic liquids as electrolytes has been reported, as they can enhance the catalytic activity of the system in terms of lowering the overpotential of the reactions and of increasing the faradaic efficiency toward a specific product.

## 2. Introduction to Ionic Liquids

Ionic liquids are substances that are entirely composed of ions and that have a melting point below 100 °C [58]. ILs have attracted attention in many applications due to their unique properties, such as thermal stability [59], chemical stability [60], conductivity [61], a relatively high CO_2_ absorption capacity [62], electrochemical stability [63], negligible volatility [64], and an ability to be used as electron transmission mediators for redox catalysis [65]. The only disadvantage of ionic liquids is their relatively high cost compared to conventional organic solvents [66]. However, the cost limitation can be minimized by using ionic liquids as additives in organic solvents as well as by recycling ionic liquids. Due to these distinguished properties, ionic liquids have recently gained attention in a variety of applications, such as in biological applications, electrolytes, separation process, heat storage, and catalysis [67]. Figure 5 shows the chemical structure of the most common IL cations and anions.

In CO_2_ electrochemical reduction, ILs have shown good catalytic performance in producing various products such as methanol (CH_3_OH) [68], carbon monoxide (CO) [45], dimethyl carbonate (C_3_H_6_O_3_) [69], formic acid (HCOOH) [38], and ethanol [70]. In this paper, the roles of the electrode, the anions, and the cations of the ionic liquids in the CO_2_ER reactions are reviewed.

## 3. The Roles of the Electrode in CO_2_ERR

Many studies have described the effect of the electrode geometry, size, interface area between the electrode and the electrolyte, and the effect of the electrical double layer on CO_2_ERR [71,72,73]. In this review, the roles of the major components are briefly highlighted, as many existing reviews have discussed the work on electrode materials extensively [74,75]. The cathode material plays a primary role of a catalyst for electroreduction in the system. Hori et al. reported the effects of different metals on CO_2_ERR, which are summarized in Figure 6 [53].

From Figure 6, it can be observed that the potential of the zinc (Zn) and indium (In) electrodes are approximately similar, but the faradaic efficiencies of the products are different. For the production of CO, the highest faradaic efficiency is 79.4% for the Zn electrode, while it is just 2.1% on the In surface. However, for HCOOH production, the highest faradaic efficiency is 94.9% on the In electrode, while it is 6.1% on the Zn surface. It can be observed that the Sn, Hg, Pb, and In metals produce HCOOH favorably, while the group of Au, Ag, Zn, and Pd metals produce CO favorably. Among all of the elements in Figure 6, copper (Cu) has the highest tendency to produce various types of products. This property makes Cu and its derivatives attractive materials for study [76,77,78]. Hansen reported that the selectivity of the product principally depends on the binding energy of CO_2_^•−^ on the metal surface [54]. He concluded that the rate in the noble metals gold (Au) and silver (Ag) is limited by CO_2_ activation because CO binds loosely. For the Pd, Ni, Pt, and Rh electrodes, CO_2_ activation and conversion to adsorbed CO is easy, and the rate is restricted by CO desorption due to massive binding. As shown in Figure 7, Cu is positioned at the center and is shown to be binding intermediately with CO and COOH compared to the other materials, which may explain the reason why a variety of products can be produced by this electrode.

It can be concluded that the electrode materials play a major role in product selectivity. Therefore, a new way to increase the catalytic activity of the electrodes is by introducing molecules on the electrode surface that can reallocate the lowest unoccupied molecular orbital (LUMO) and the highest occupied molecular orbital (HOMO), as reported by Yadav and Singh [79].

## 4. Ionic Liquids as Co-Catalyst for CO_2_ERR System

Many studies have been performed to elucidate the role of ILs in CO_2_ERR. Table 2 shows a non-exhaustive list of ionic liquids used in CO_2_ERR, with the list of IL abbreviations being shown in Table 3. Although various ionic liquids have been reported, the role of ILs and the exact mechanism behind the reactions remain an open question. The ionic liquids that are incorporated into the liquid electrolyte aid in increasing CO_2_ solubility while enhancing the catalytic system at the same time. As ionic liquids are composed of cations and anions, the effects of each component are discussed separately for a better understanding.

### 4.1. The Effect of Anions on the CO_2_ERR System

Many research studies have reported on the solubility of CO_2_ in ILs using prediction and experimental methods [92,93,94,95,96,97]. Aki and Mellein [89] studied the effect of anions on CO_2_ solubility, where they used seven ILs with 1-butyl-3-methylimidazolium ([Bmim]) as the cation and different anions namely bis(trifluoromethylsulfonyl)imide ([Tf_2_N]), tetrafluoroborate ([BF_4_]), tris(trifluoromethylsulfonyl)methide ([methide]), nitrate ([NO_3_]), trifluoromethanesulfonate ([OTf]), dicyanamide ([DCA]), and hexafluorophosphate ([PF_6_]). They also investigated the effect of alkyl chain length on the cation using 1-octyl-3-methylimidazolium bis(trifluoromethylsulfonyl)imide ([Omim][Tf_2_N]), 1-hexyl-3-methylimidazolium bis(trifluoromethylsulfonyl)imide ([Hmim][Tf_2_N]), and 2,3-dimethyl-1-hexylimidazolium bis(trifluoromethylsulfonyl)imide ([DMHxIm][Tf_2_N]). The results show that the solubility of carbon dioxide is mainly dependent on the anions, and it is higher in anions that contain fluoroalkyl groups such as [Tf_2_N] and [methide]. They attributed this to the acid/base interactions of CO_2_ with anions. They also noted that the CO_2_ solubility slightly increases when the alkyl chain length of the cation increases due to a higher free volume in the ILs with longer alkyl chains.

On the other hand, Anthony and Anderson [98] studied the solubility of various gases (carbon monoxide, oxygen, carbon dioxide, ethylene, ethane, nitrous oxide, and benzene) in butyl-methyl pyrrolidinium bis(trifluoromethylsulfonyl) imide, tri-isobutyl-methyl phosphonium p-toluenesulfonate, and methyl-tributylammonium bis(trifluoromethylsulfonyl) imide, where they observed that carbon dioxide and nitrous oxide have strong interactions with ILs followed by interactions with ethylene and ethane. However, oxygen has demonstrated very poor solubility and limited interaction with ILs. For carbon monoxide, the authors could not detect the solubility because of limitations with their apparatus. They concluded that the ILs with the bis(trifluoromethylsulfonyl) imide anion had the highest CO_2_ solubility, regardless of whether the cation was tetraalkylammonium, pyrrolidinium, or imidazolium. Their study highlighted that anions have the most significant effect on gas solubilities. Similarly, Almantariotis and Stevanovic studied the absorption of carbon dioxide, nitrogen, ethane, and nitrous oxide by 1-alkyl-3-methylimidazolium (C_n_mim, n = 2,4,6) tris(pentafluoroethyl)trifluorophosphate ionic liquids (FAP). They observed that the ILs containing highly fluorinated anions (tris(pentafluoroethyl)trifluorophosphate [FAP]) recorded the highest CO_2_ solubility among the ILs for the same cations [91]. They noted that amino acid-based ILs have high CO_2_ solubility due to the interaction between carbon dioxide and amino functional groups [99,100].

Snuffin and Whaley [90] observed that the electroreduction of CO_2_ in 1-butyl-3-methylimidazolium tetrafluoroborate [Bmim][BF_4_] and 1-butyl-3-methylimidazolium bis(trifluormethylsulfonyl)imide [Bmim][TF_2_N] is ineffective compared to [Emim][BF_3_Cl], which has a C_3_-symmetric tetrahedral structure. They noted that the dative B-Cl covalent bond is not as strong as the ordinary B-F covalent bond. The B-Cl bond-length was 0.173 nm, which is 33% greater than the B-F bond length of 0.130 nm. They reported that the dissolved CO_2_ molecules may replace the C_l_ atoms in the BF_3_Cl by forming BF_3_-CO_2_ adducts. On the other hand, the oxygen atoms of CO_2_ are Lewis bases; therefore, the BF_3_ can form a Lewis acid–base adduct with CO_2_. They studied [Emim][BF_3_Cl] IL as an electrolyte for the electrochemical reduction of CO_2_ at 1 atm and 25 °C on a platinum electrode and discovered that the reaction occurred at −1.8 V vs. Ag wire, which is less negative than previously reported research, where the reaction occurred at −2.4 V vs. Ag/AgCl in 1-butyl-3-methylimidazolium tetrafluoroborate [Bmim][BF_4_] on a Cu electrode and at −2.0 V vs. Ag wire at 50 °C in [Bmim][BF_4_] on a Ag electrode. The current density was as high as −5.7 mA/cm^2^ [101,102,103]. They explained that the increase in the catalysis performance occurred through the reduction of the overpotential, which was due to the formation of the Lewis acid–base adduct BF_3_-CO_2_. It can be concluded that the anion affects both the catalytic and solubility; however, its highest effect is on the solubility, whereas the influence in reducing the overpotential is less important because of its negative charge, making it far apart from the cathode.

### 4.2. The Effect of the Cations on the Activity of CO_2_ERR System

The low adsorption of CO_2_ on the electrode surface and the linear stability of the CO_2_ structure and its hybridization are the reason behind the difficulty in activating CO_2_. Thus, a high overpotential is required to reduce the large energy barrier for CO_2_^•−^ radical initiation, which is considered to be the reaction-determining step. More attention has been given to the cations since they have been reported as the central active part of catalytic performance [87].

Vasilyev et al. [104] summarized the role of imidazolium-based cations and highlighted that the obtained results are diverse and conflicting and that they can be divided into two major groups. The first group suggests that a covalent bond is formed between the IL and CO_2_ (as shown in Figure 8A, carboxylic adduct formation) or that there is a hydride transfer (Figure 8B); the second assumption is based on non-covalent interactions (stabilization of the complex through hydrogen bonds or changes in the local CO_2_ environment, as seen in Figure 8C,D, respectively). We observed that this diversity in the results is due to the CO_2_ absorption type (physical or chemical), which is determined by the dominant molecular force between the CO_2_ and the imidazolium-based group.

Rosen et al. and Masel performed many experiments to understand the mechanism of CO_2_ERR [45,105,106,107]. They observed that ILs have a good ability to reduce the overpotential (i.e., merely 200 mV higher than the equilibrium potential) with a high faradaic efficiency of 96% on the silver cathode (Ag) [45]. Moreover, the authors reported that the ILs showed a high ability to suppress the competing H_2_ generation reaction by building a monolayer on the electrode. They highlighted that the [Emim] cations are mostly located at the metal catalyst surface of the electrode throughout the electrolysis and suggested that CO_2_ is not directly adsorbed onto the metal catalyst surface of the electrode but that it is the [Emim] bound to the CO_2_ complex ([Emim]-CO_2_) that is adsorbed on the electrode surface, which is the reason for the enhancement observed in the catalytic activity in terms of lowering the overpotential, as illustrated in Figure 9. They also noted that the addition of water into the IL promotes catalytic performance, which shows that the hydrogen bond interaction promotes the catalyst efficiency. By comparing their results with those obtained in the study by Li et al. [108], who concluded that the addition of water narrows the electrochemical window of ionic liquids, we can observe that the addition of water in Rosen’s experiments also narrows the electrochemical stability of the electrolyte by increasing the reduction potential. It should be noted that the oxidation potential (loss of electrons) or reduction potential (gain of electrons) of a molecule is closely related to the ionization energy, which is the amount of energy that is required to remove an electron from the highest occupied molecular orbital (HOMO) and the electron affinity, which is the amount of energy that is released when an electron is added to the lowest unoccupied molecular orbital (LUMO), respectively. This decreases the LUMO values since the reduction is related to the LUMO energy level, whereby the HOMO and LUMO values can be approximated using Equation (1) and Equation (2), respectively [109]:E_HOMO_ = −(E_(ox)_onset + 4.8 − E_FOC_) eV (1)
E_LUMO_ = −(E(_red)_onset + 4.8 − E_FOC_) eV(2)
where E_HOMO_ and E_LUMO_ correspond to the energy levels of HOMO and LUMO, E_ox(onset)_ is the onset of the oxidation potential, E_red(onset)_ is the onset of the reduction potential, 4.8 is the reference energy fpr ferrocene (4.8 eV under the vacuum level), and E_FOC_ is the potential of ferrocene/ferrocenium (FOC/FOC^+^) versus Ag/AgCl. Based on this approximation, it can be deduced that the LUMO energy values could be the reason behind the electroreduction reactivity. Moreover, the interaction between the OH^−^ anion from the added water in the cathode side and the H^+^ from the anode side can obstruct the competing H_2_ generation reaction through the attraction force, which suppresses the competitive HER.

Sun et al. and Ramesha [85] studied the selectivity of the reaction towards CO, where they used Pb as the cathode and [Emim][Tf_2_N] as the electrolyte, and they found that the production of CO can be attributed to the interaction between CO_2_ and the cation [Emim]^+^ monolayer at the cathode surface. The results indicated that the CO_2_^•−^ and [Emim]^+^ adsorbed on the cathode surface could direct the CO_2_^•−^ reaction toward forming CO rather than oxalate. They concluded that [Emim]^+^ stabilized the CO_2_^•−^ radical, which shifted the reaction in the direction of forming CO rather than oxalate, as shown in Figure 10. Hu et al. studied CO_2_ reduction by using MoO_2_ as the cathode and different imidazolium-based ILs as electrolytes at several temperatures and pressures [86]. The IL 1-butyl-3-methylimidazolium hexafluorophosphate [Bmim][PF_6_] recorded the highest activity results at low temperatures. They demonstrated that the IL cations play the role of the co-catalyst in enhancing catalytic activity.

Several studies [110,111] have been conducted to enhance catalysis performance. Nonetheless, the design of a high-performance catalysis system remains a challenge. Many researchers [112,113,114,115] have studied the thin area between the electrode and the electrolyte where the electrochemical reaction takes place (electrical double layer region). They concluded that the electrical double layer could be the reason behind the problem of the system having a high over potential.

Lau and Vasilyev [87] studied the different structures of imidazolium-based ILs supported by a 0.1 M solution of tetrabutylammonium hexafluorophosphate [TBA][PF_6_] in anhydrous acetonitrile. They evaluated the performance of different structures as co-catalysts for the CO_2_ERR on a silver (Ag) electrode. It was reported that the catalytic activity mainly originates in the cation and that protons at C_4_ and C_5_ are critical, as shown in Figure 11. We deduced that this is due to the π-bond, which enhances the acidity and therefore also enhances the acid/base interaction between CO_2_ (which is relatively considered to be a Lewis base) and the cations (which is considered to be a Lewis acid).

Vasilyev and Shirzadi [88] investigated the CO_2_ reduction in pyrazolium ILs (Pz-IL) on a silver electrode, where they observed a significant decrease in the onset potential for the reduction (approximately 500 mV). They proceeded with the electrochemical conversion in acetonitrile-based electrolytes containing Pz-IL as co-catalysts with faradaic efficiencies (FE) of nearly 100% for CO over a range of at least 0.5 V. The molecular structures of the pyrazolium that were used are shown in Figure 12, and the potentials for the different structures are shown in Figure 13. They suggested that the co-catalyst effect comes from ion pair stabilization, ion interactions, and modifications in the local microenvironment. They reported that the nitrogen domain in the pyrazolium cation is the most charged region and that substitutions at the C_3_ and C_5_ positions change the system more than when substitutions are made at the C_4_ position. They assumed that the CO_2_^•−^ intermediate interacts with the positively charged region of the Pz ring, which helps to stabilize the intermediate and to reduce the reaction overpotential.

The high stability of CO_2_ comes from its molecular orbital, as shown in Figure 14. The molecular orbitals (MO) with the highest relevance to the reactivity are the 1π_g_ and 2π_u_ orbitals, which play the role of HOMO and LUMO, respectively. The doubly occupied nonbonding 1π_g_ MOs are mainly localized at the terminal oxygen atoms, whereas the empty antibonding 2π_u_ orbitals are mostly centered on the carbon atom. Therefore, CO_2_ can be considered to be an amphoteric molecule, where the oxygen atoms display a Lewis basic character and where the carbon atom behaves in a similar manner to a Lewis acid center with a slightly negative electron affinity (Ea) of about −0.6 eV [116] and a first ionization potential (I_P_) of about 13.8 eV [117]. Carbon dioxide is more of an electron acceptor than it is an electron donor. The molecule’s reactivity is regulated by the electrophilic characteristics of the carbon atoms rather than the weak nucleophilic properties of the oxygen atoms. The resulting lowest energy state corresponds to a curve geometry when the LUMO orbitals of CO_2_ are occupied with electrons. The CO_2_^•−^radical anion, for example, is a twisted molecule with a balanced angle of 134° [118]. Any carbon dioxide interaction with an electron-rich metalcore would also induce a loss of linearity [119].

Buijs and Witkamp [122] used quantum chemical calculations to predict the electrochemical window of ILs with reduction-resistant anions. They conducted their experiments using the Spartan’10 molecular modeling program suite. They optimized the structures at the B3LYP level using the corresponding PM3 structures as input. They highlighted that the lowest unoccupied molecular orbital (LUMO) of the ILs containing reduction-resistant anions is entirely concentrated on the cation. They concluded that the order in resistance against reduction is piperidinium > pyrrolidinium > quaternary phosphonium > quaternary ammonium > imidazolium > pyrazolium pyridinium [122]. This sequence is consistent with the experimental data [123,124,125].

From this review, by comparing the performance of imidazolium-based ILs [87] with the pyrazolium-based ILs [88] and by taking the lowest unoccupied molecular orbital (LUMO) values [122] into consideration, it can be stipulated that the cation structures with the lowest LUMO energy levels have a higher hydrogen bond interaction value [126,127] and higher catalytic efficiency in terms of reducing the overpotential. Since the complex of (CO_2_^•−^ cation) formed is adsorbed on the catalyst, the CO_2_ properties could change in terms of the LUMO energy levels, which may be due to the acid/base interaction between the CO_2_ (which is relatively considered to be a Lewis base) and the cations (which is considered to be a Lewis acid). The other possibility is that due to the low LUMO value, the high hydrogen bonding energy can disturb the strong linearity of the CO_2_ and can decrease the strength of the covalent bond. Consequently, this will reduce the activation energy that is required and will reduce the overpotential. To determine which hypothesis is more dominant, a study considering the type of CO_2_ absorption (physical or chemical) is recommended.

It is noteworthy to highlight the effect of the alkyl chain length of the cations on the solubility, as reported by Safavi and Ghotbi [128]. They studied the solubility of carbon dioxide and hydrogen sulfide in the 1-alkyl-3-methylimidazolium hexafluorophosphate ([C_n_mim][PF_6_]) IL at different temperatures and pressures. They found that when the cation alkyl chain length becomes longer, the solubility of CO_2_ and H_2_S in the ionic liquid increases. It can be concluded that the reason behind the improvement of the solubility is due to the increase in the non-polarity since CO_2_ is nonpolar, thus behaving in the same way that it would when dissolving. They also concluded that the solubility of carbon dioxide increased when the pressure increased and decreased when the temperature increased. The high temperature reduces the risk of overpotential because of the higher activation energy, it but decreases the solubility of carbon dioxide, which is not very high (0.033 mol/dm^3^ at standard temperature, 1 atm and pH = 7) in aqueous solutions [129]. Moreover, the reaction components that are adsorbed on the electrode are also lowered. The use of an organic solvent can help to solve this problem in terms of increasing the solubility of CO_2_ [130]. In non-protic solvents, an increase in the temperature changes the products of CO_2_ reduction, and CO is favored over oxalate formation at higher temperatures [131].

From this review, ionic liquids have good potential as co-catalyst for CO_2_ER because of the numerous possible combinations between the cations and anions that lead to a diverse suite of characteristics that serve the application. For instance, in CO_2_ electrochemical reduction, the cations must have low LUMO values while also maintaining the electrochemical stability of the ILs. This can be achieved through molecular interaction by selecting the suitable types of anions [132].

## 5. Conclusions

For CO_2_ERR, ionic liquids play an essential role due to their unique physical chemistry properties, CO_2_^•−^stabilization, their ability to lower the overpotential, and their ability to increase the faradaic efficiency and current density. The main role of IL anions is to exhibit high CO_2_ absorption on the catalysis surface as well as a co-catalyst in terms of reducing the overpotential if it forms a Lewis acid–base adduct. The IL cation that is closer to the electrode plays the main role as a co-catalyst in terms of reducing overpotential, and the formation of the cation-CO_2_ complex could change the properties of the system in terms of the hydrogen bond interaction energy, the electrical double layer region, and the relocation of the HOMO and LUMO values. Most of the research highlighted that the strong hydrogen interaction could be the reason why the catalytic efficiency is enhanced; however, since the LUMO values and hydrogen interaction energy are related, we concluded that the main reason for this could be the relocation of the LUMO values. Moreover, we highlighted that the determination of the nature of the absorption (physical/chemical) of CO_2_ in the ionic liquids is required to further understand the mechanism. Cations also play a secondary role in increasing the solubility of CO_2_ by increasing the alkyl chain length. The material of the electrode plays the main role of product selectivity, according to the binding energy between the electrode and CO_2_ radical. Furthermore, the catalytic activity of the electrode could be enhanced by introducing molecules on the electrode surface that could reallocate the lowest unoccupied molecular orbital and increase the catalytic performance.

## Figures and Tables

**Figure 1 molecules-26-06962-f001:**
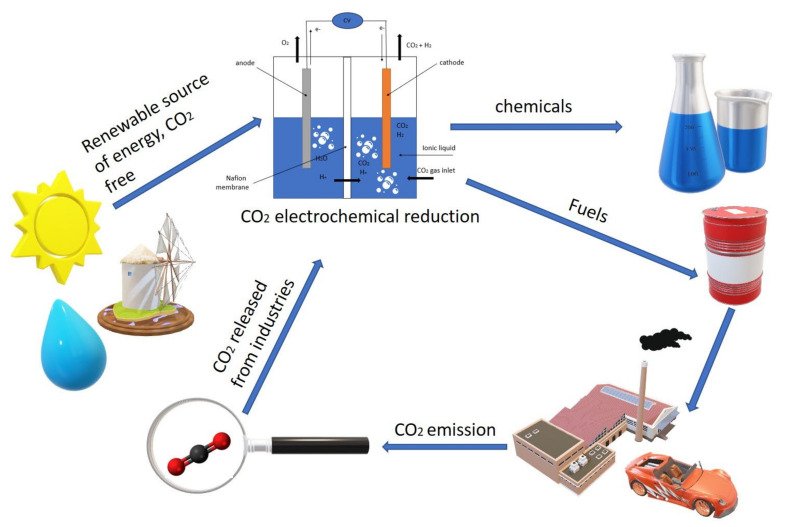
Carbon dioxide reduction cycle using renewable and green source of energy.

**Figure 2 molecules-26-06962-f002:**
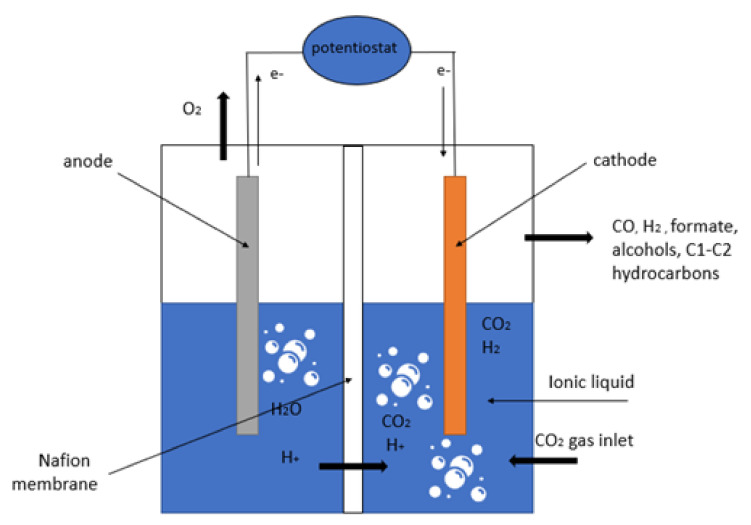
Illustration of an electrochemical H-cell for CO_2_ reduction.

**Figure 3 molecules-26-06962-f003:**
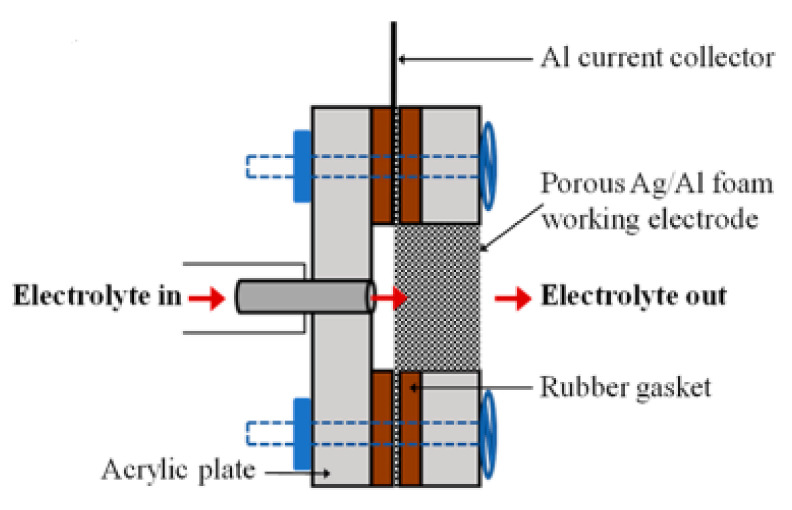
Illustration of the flow cell for CO_2_ reduction. Reprinted with permission from [20]. Copyright 2014, American Chemical Society.

**Figure 4 molecules-26-06962-f004:**
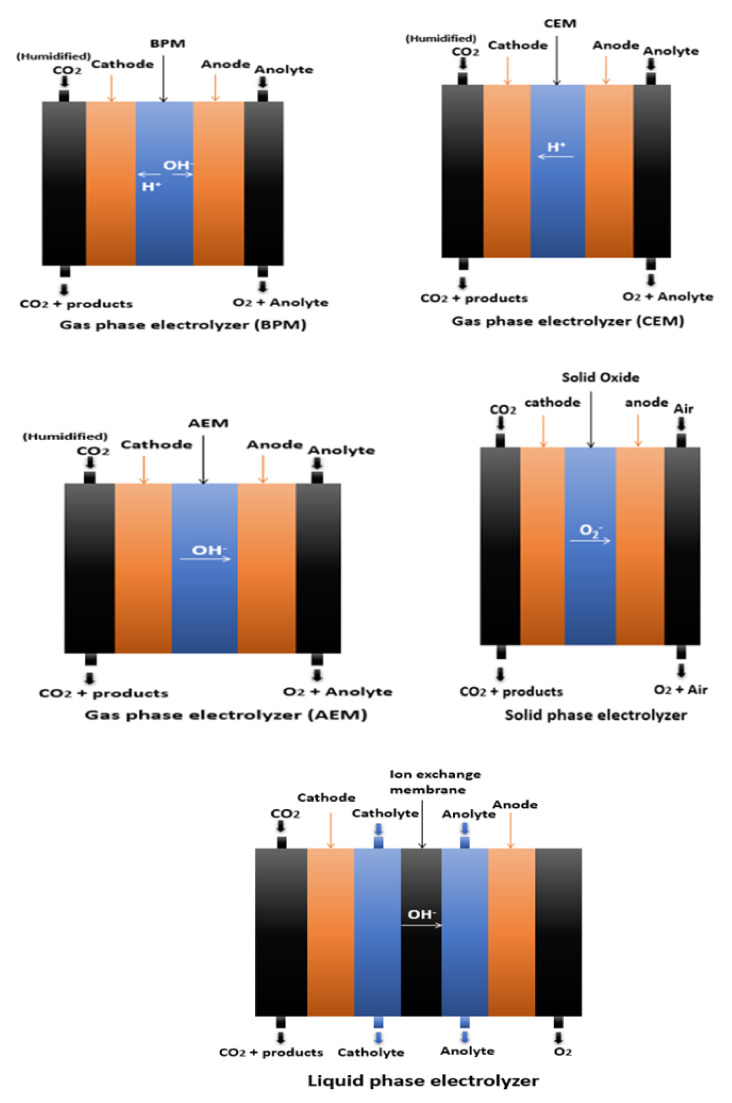
Different electrolyzer configurations for CO_2_ERR [18].

**Figure 5 molecules-26-06962-f005:**
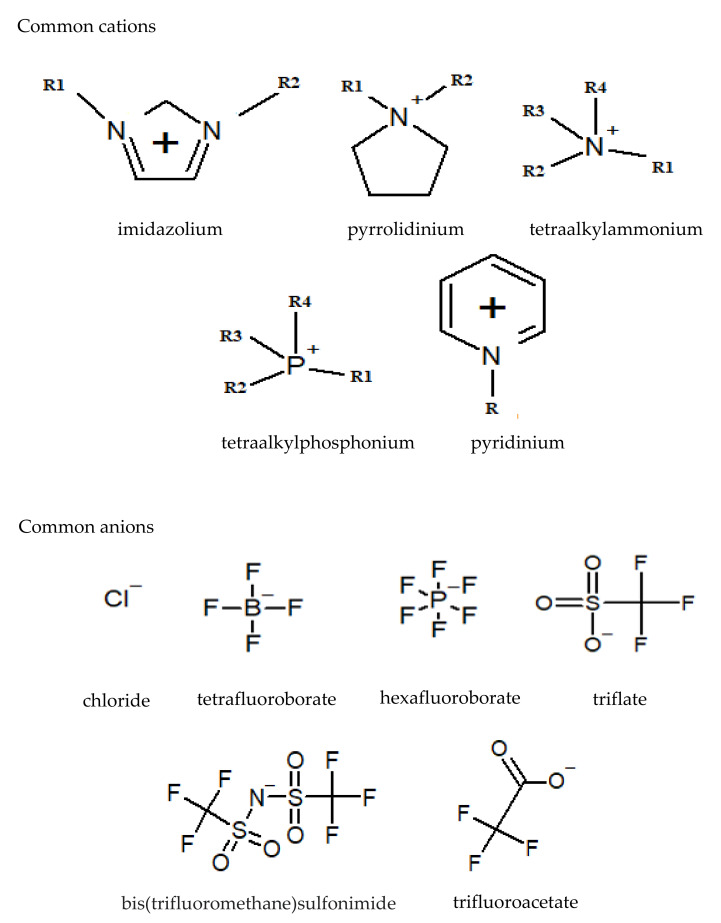
Chemical structure of the most common IL cations and anions.

**Figure 6 molecules-26-06962-f006:**
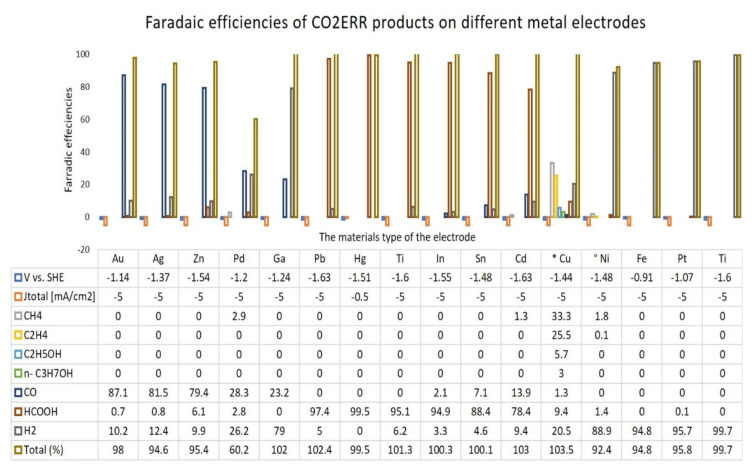
Faradaic efficiencies of CO_2_ERR products on different metal electrodes in 0.1 M KHCO_3_ solution at 18.15 ± 0.05 °C saturated with carbon dioxide. * The total value contains C_3_H_5_OH (1.4%), CH_3_CHO (1.1%), andC_2_H_5_CHO (2.3%) in addition to the tabulated substances. The total value contains C_2_H_6_ (0.2%) [53].

**Figure 7 molecules-26-06962-f007:**
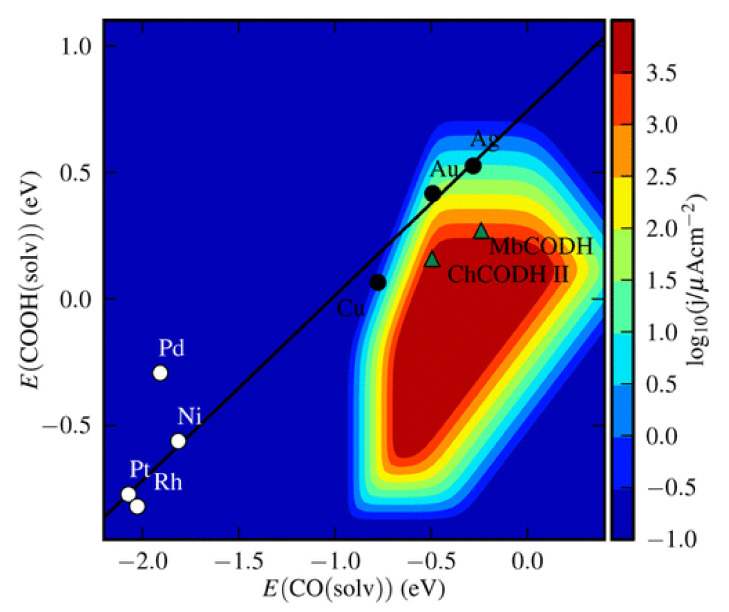
Kinetic volcano at 0.35 V overpotential for CO propagation from the transition metal (211) stage. The transition metals follow a linear trend that does not cross over the peak of the volcano. The noble metals, on the other hand, reach the trend line’s optimum. The specific CO generation current from the ChCODH II and MbCODH enzyme models is comparable to, if not superior to, that of the noble metals. Reprinted with permission from [54]. Copyright 2014, American Chemical Society.

**Figure 8 molecules-26-06962-f008:**
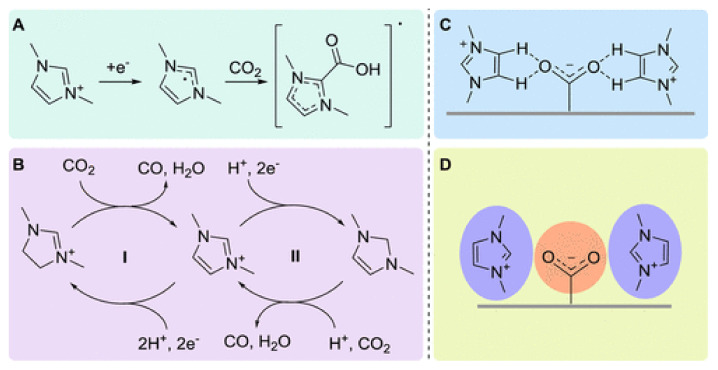
Mechanisms proposed regarding the role of IL as co-catalysts in the CO_2_RR. Left: covalent activation. (**A**) Generation of a covalent bond between CO_2_ and the imidazolium IL. (**B**) Im IL as a hydride/proton donor reducing CO_2_. Right: non-covalent activation. (**C**) Stabilization of CO_2_^•−^ by hydrogen bonding. (**D**) Stabilization of CO_2_^−^ through the adjustment of the local electric field. Reprinted with permission from [104]. Copyright 2014, American Chemical Society.

**Figure 9 molecules-26-06962-f009:**
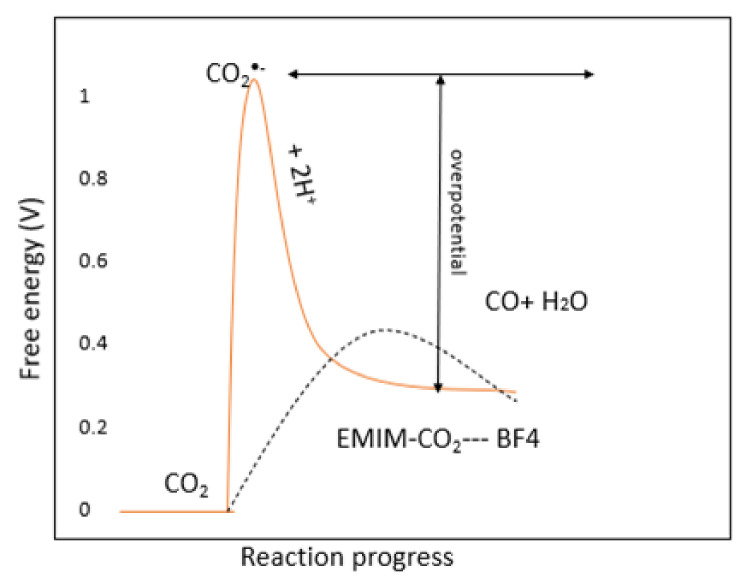
A schematic of how the free energy of the system changes during the reaction of CO_2_ + 2H^+^ + 2e^−^ → CO + H_2_O in water or acetonitrile (solid line) or in the IL [Emim][BF_4_] (dashed line) [45].

**Figure 10 molecules-26-06962-f010:**
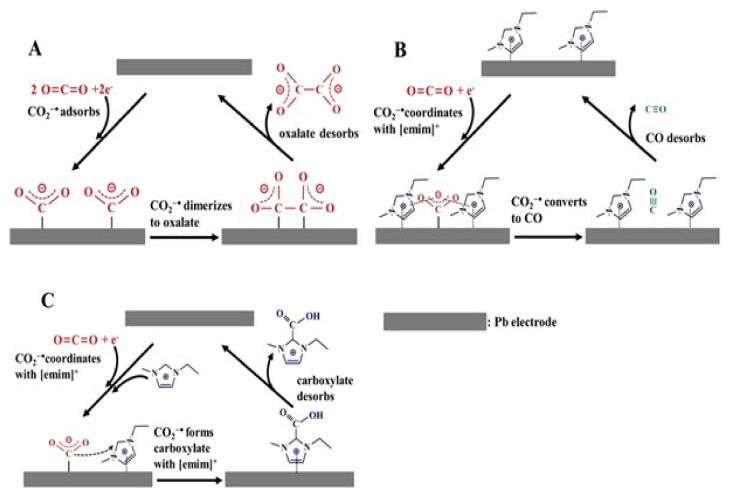
Reaction pathways for the electrochemical reduction of CO_2_ in (**A**) the absence and (**B**,**C**) presence of [Emim][Tf_2_N] at a Pb electrode in acetonitrile. Reprinted with permission from [85]. Copyright 2014, American Chemical Society.

**Figure 11 molecules-26-06962-f011:**
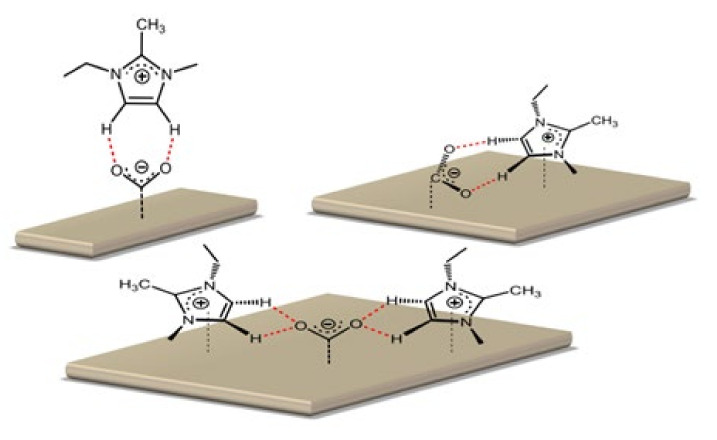
The interaction between the active imidazolium protons and CO_2_. Reprinted with permission from [87]. Copyright 2014, American Chemical Society.

**Figure 12 molecules-26-06962-f012:**
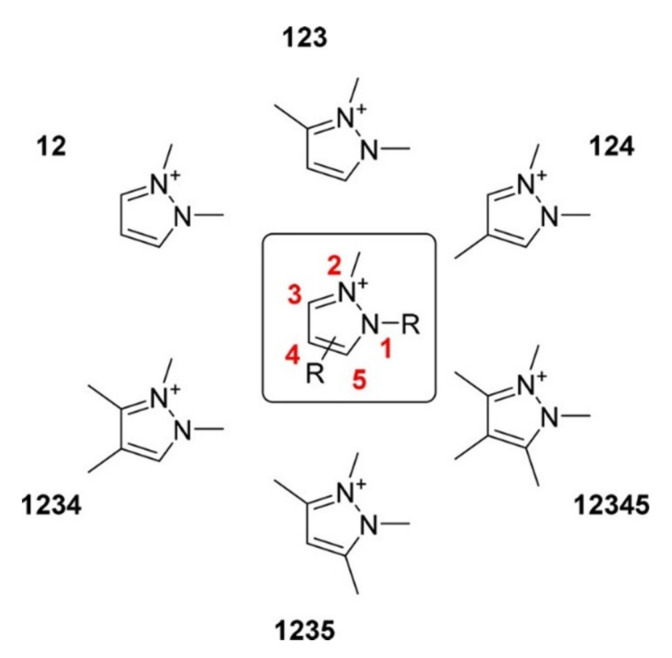
Pyrazolium cations used in the Vasilyev study [88]. Reprinted with permission from [88]. Copyright 2018, American Chemical Society.

**Figure 13 molecules-26-06962-f013:**
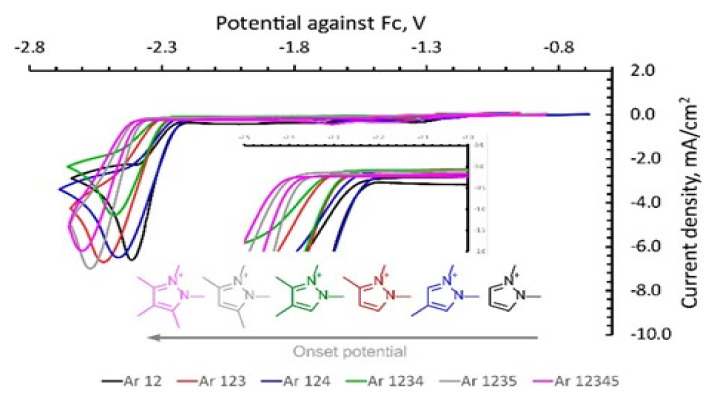
The CV values to illustrate the differences in the stability of several substituted Pz cations under Ar (Ag polished disk electrode, 0.1 M NBu_4_PF_6_ in dry acetonitrile, and 0.02 M IL additive). Reprinted with permission from [88]. Copyright 2018 Chemical Society.

**Figure 14 molecules-26-06962-f014:**
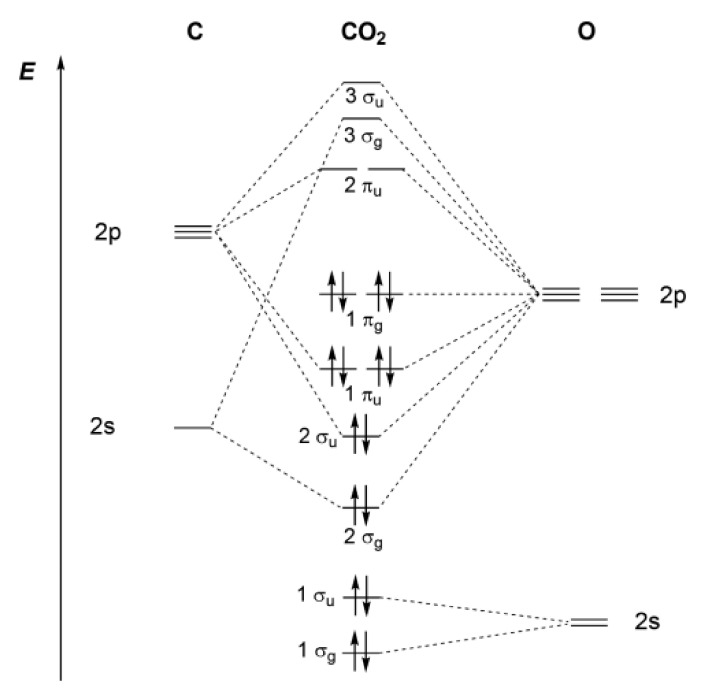
Qualitative molecular orbital diagram of carbon [16,120,121]. Reprinted with permission from [16]. Copyright 2018, American Chemical Society.

**Table 1 molecules-26-06962-t001:** Products of CO_2_ERR with equilibrium potentials. Reprinted with permission from [41] 2019, American Chemical Society.

Reaction	E_0_/(V vs. Reversible Hydrogen Electrode RHE)	(Product) Name, Abbreviation
2H_2_O → O_2_ + 4H^+^ + 4e^−^	1.23	Oxygen Evolution Reaction, OER
2H^+^ + 2e^−^ → 2H_2_	0	Hydrogen Evolution Reaction, HER
xCO_2_+ nH^+^ + ne^−^ →product + yH_2_O		CO_2_ Reduction, CO_2_R
CO_2_ + 2H^+^ + 2e^−^ → HCOOH (aq)	−0.12	Formic acid
CO_2_ + 2H^+^ + 2e− → CO(g) + H_2_O	−0.10	Carbon monoxide
CO_2_ + 6H^+^ + 6e^−^ → CH_3_OH (aq) + H_2_O	0.03	Methanol, MeOH
CO_2_ + 4H^+^ + 4e^−^ → C(s) + 2H_2_O	0.21	Graphite
CO_2_ + 8H^+^ + 8e^−^ → CH_4_(g) + 2H_2_O	0.17	Methane
2CO_2_ + 2H^+^ + 2e^−^ → (COOH)_2_ (s)	−0.47	Oxalic acid
2CO_2_ + 8H^+^ + 8e^−^ → CH_3_COOH (aq) + 2H_2_O	0.11	Acetic acid
2CO_2_ + 10H^+^ + 10e^−^ → CH_3_CHO (aq) + 3H_2_O	0.06	Acetaldehyde
2CO_2_ + 12H^+^ + 12e^−^ → C_2_H_5_OH (aq) + 3H_2_O	0.09	Ethanol, EtOH
2CO_2_ + 12H^+^ +12e^−^→ C_2_H4(g) + 4H_2_O	0.08	Ethylene
2CO_2_+ 14H^+^ + 14e^−^→ C_2_H_6_(g) + 4 H_2_O	0.14	Ethane
3CO_2_+ 16H^+^ + 16e^−^ → C_2_H_5_CHO (aq)+ 5H_2_O	0.09	Propionaldehyde
3CO_2_ + 18H^+^ + 18e^−^ → C_3_H_7_OH (aq) + 5H_2_O	0.10	Propanol, PrOH
xCO + nH^+^ + ne^−^ → product + yH_2_O		CO Reduction, COR
CO+ 6H^+^ + 6e^−^→ CH_4_(_g_) + H_2_O	0.26	Methane
2CO+ 8H^+^ + 8e^−^ → CH_3_CH_2_OH (aq) +H_2_O	0.19	Ethanol, EtOH
2CO + 8H^+^ + 8e^−^ → C_2_H_4_(g) + 2H_2_O	0.17	Ethylene

**Table 2 molecules-26-06962-t002:** Some examples of ionic liquids used in CO_2_ER.

Electrode	Electrolyte	Potential	Faradaic Efficiency of the Product (%)	Current Density (mA⋅cm^−2^)	Reference
Sn	40 mg·mL^−1^ [C_3_mim] OTf aqueous solution	−2.1 V (vs. Ag/AgCl)	HCOOH (35.47)	9.3	[80]
In	40 mg·mL^−1^ [C_3_mim] OTf aqueous solution	−2.1 V (vs. Ag/AgCl)	HCOOH (73.90)	11.9	[80]
Pb phytate	12.8 wt% [Bmim]BF_4_/9.9 wt% H_2_O/acetonitrile	−2.25 V (Vs. Ag/Ag^+^)	HCOOH (92.7)	30.5	[81]
Pb	4-(methoxycarbonyl) phenol tetraethylammonium ([TEA][4-MF-PhO])	−2.6 V (vs. Ag/Ag^+^)	H_2_C_2_O_4_ (86)	9.03	[82]
Cu nanoporous foam	0.04 mol/L[Bmim][Br] + 0.1 mol/L KHCO_3_	−1.6 vs. Ag/Ag	Ethanol (49)	20	[70]
Au	0.1 M [Bmim][OAc] + 0.2 vol% H_2_OinDMSO	−1.8 vs. Ag/Ag	CO (98)	8.6	[83]
Ag	[Bmim][Cl] with 20 wt% H_2_O	−1.5 vs. SCE	CO > 99	2.4	[84]
Ag	[Emim]BF_4_/18% water	−1.50 vs. cell potential	CO (96)	n/a	[45]
Pb	0.1 M [Emim][Tf_2_N]/AcN	−2.25 V (vs. Ag/AgNO_3_)	CO (44) Carboxylate(74)	-	[85]
Pb	0.1 M TEAP/AcN	−2.40 V (vs. Ag/AgNO_3_)	CO (10) Oxalate(70)	-	[85]
MoO2/Pb	0.3M [Bmim]PF_6_/MeCN/0.1 M H_2_O	−2.45 V (vs. Fc/Fc^+^)	H_2_ (12.4) CO(60.8) C_2_O_4_^−2^(5.3) HCOO^−^(17.8)	-	[86]
MoO2/Pb	0.3M [Bmim]PF_6_/MeCN/0.2 M H_2_O	−2.45 V (vs. Fc/Fc^+^)	H_2_ (25.1) CO(51.7) C_2_O_4_^−2^(5.5) HCOO^−^(9.8)	-	[86]
MoO2/Pb	0.3M [Bmim]PF_6_/MeCN/0.3 M H_2_O	−2.45 V (vs. Fc/Fc^+^)	H_2_ (28.9) CO(51.4) C_2_O_4_^−2^(4.3) HCOO^−^(6.2)	-	[86]
Ag	0.02 M [1a][BF_4_]/0.1 M NBu_4_PF_6_ in dry acetonitrile	−2.46 V (vs. Ag/AgCl)	CO(99.8)	4.2	[87]
Ag	0.02 M [2a][BF_4_]/0.1 M NBu_4_PF_6_ in dry acetonitrile	−2.37 V (vs. Ag/AgCl)	CO(100)	4.2	[87]
Ag	0.02 M [3a][BF_4_]/0.1 M NBu_4_PF_6_ in dry acetonitrile	−2.34 V (vs. Ag/AgCl)	CO(100)	2.8	[87]
Ag	0.02 M [4a][BF_4_]/0.1 M NBu_4_PF_6_ in dry acetonitrile	−2.35 V (vs. Ag/AgCl)	CO(99)	4.2	[87]
Ag	0.02 M [5a][BF_4_]/0.1 M NBu_4_PF_6_ in dry acetonitrile	−2.43 V (vs. Ag/AgCl)	CO(100)	4.2	[87]
Ag	[Bmim][BF_4_}	−2.11 V (vs. Fc/Fc^+^)	CO(92) H_2_(<1)	-	[88]
Ag	0.02 M [Pz1235B] [TFSI] /0.1 M NBu_4_PF_6_ in dry acetonitrile	−2.11 V (vs. Fc/Fc^+^)	CO(99) H_2_(4) CH_4_(0.9)	-	[88]

**Table 3 molecules-26-06962-t003:** List of ionic liquid abbreviations.

Name	Abbreviation	Reference
1-butyl-3-methylimidazolium bis(trifluoromethylsulfonyl)imide	[Bmim][Tf_2_N]	[89,90]
1-butyl-3-methylimidazolium tetrafluoroborate	[Bmim][BF_4_]	[89,90]
1-butyl-3-methylimidazolium tris(trifluoromethylsulfonyl)methide	[Bmim][methide]	[89]
1-butyl-3-methylimidazolium nitrate	[Bmim][NO_3_]	[89]
1-butyl-3-methylimidazolium trifluoromethanesulfonate	[Bmim][OTf]	[89]
1-butyl-3-methylimidazolium dicyanamide	[Bmim][DCA]	[89]
1-butyl-3-methylimidazolium hexafluorophosphate	[Bmim][PF_6_]	[89]
1-octyl-3-methylimidazolium bis(trifluoromethylsulfonyl)imide	[Omim][Tf_2_N]	[89]
1-hexyl-3-methylimidazolium bis(trifluoromethylsulfonyl)imide	[Hmim][Tf_2_N]	[89]
2,3-dimethyl-1-hexylimidazolium bis(trifluoromethylsulfonyl)imide	[DMHxIm][Tf_2_N]	[89]
1-ethyl-3-methylimidazolium tris(pentafluoroethyl)trifluorophosphate ionic liquids (FAP)	[Emim][FAP]	[91]
1- butyl -3-methylimidazolium tris(pentafluoroethyl)trifluorophosphate ionic liquids (FAP)	[Bmim][FAP]	[91]
1- hexyl -3-methylimidazolium tris(pentafluoroethyl)trifluorophosphate	[Hmim][FAP]	[91]
1-ethyl-3-methylimidazolium trifluorochloroborate	[Emim][BF_3_Cl]	[90]
1-ethyl-3-methylimidazolium tetrafluoroborate	Emim BF_4_	[45]
1-butyl-3-methylimidazolium hexafluorophosphate	[Bmim][PF_6_]	[86]
1-ethyl-3-methylimidazolium bis(trifluoromethylsulfonyl)imide	[Emim][Tf_2_N]	[85]
1,2-dimethylpyrazolium bis(trifluoromethylsulfonyl)imide	[Pz12][Tf_2_N]	[88]
1,2,3-trimethylpyrazolium bis(trifluoromethylsulfonyl)imide	[Pz123][Tf_2_N]	[88]
1,2,4-trimethylpyrazolium bis(trifluoromethylsulfonyl)imide	[Pz124][Tf_2_N]	[88]
1,2,3,4,5-pentamethylpyrazolium bis(trifluoromethylsulfonyl)imide	[Pz12345][Tf_2_N]	[88]
1,2,3,5-tetramethylpyrazolium bis(trifluoromethylsulfonyl)imide	[Pz1235][Tf_2_N]	[88]
1,2,3,4-tetramethylpyrazolium bis(trifluoromethylsulfonyl)imide	[Pz1234][Tf_2_N]	[88]
Tetrabutylammonium hexafluorophosphate	NBu_4_PF_6_	[88]
3-ethyl-1-methyl-1H-imidazolium tetrafluoroborate	[1a][BF_4_]	[87]
3-ethyl-1,2-dimethyl-imidazolium tetrafluoroborate	[2a][BF_4_]	[87]
3-ethyl-1,2,4,5-tetramethyl-imidazolium tetrafluoroborate	[3a][BF_4_]	[87]
1,3-dimethyl-2-phenyl-1H-imidazolium tetrafluoroborate	[4a][BF_4_]	[87]
1,3-dimethyl-2-(4-methoxyphenyl)-imidazolium tetrafluoroborate	[5a][BF_4_]	[87]
1-butyl-2,3,5-trimethyl-pyrazolium	[Pz1235B] [TFSI]	[88]

## Data Availability

Not applicable.

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
