# Peer review of "Elucidation of the Roles of Ionic Liquid in CO2 Electrochemical Reduction to Value-Added Chemicals and Fuels"

_molecules, 2021, doi:10.3390/molecules26226962_

Round 1
Reviewer 1 Report
The manuscript titled “Elucidation of the roles of ionic liquid in CO2 electrochemical reduction to value-added chemicals and fuels” authored by Yahya et al. reviewed the role of ionic liquid in CO2 electrochemical reduction. This manuscript suffers from the following points:
- As suggested by the title and abstract that this manuscript focuses on ionic liquids, the general knowledge about CO2 electroreduction (types of reactors, types of products) should focus more on works using ionic liquids to be more relevant.
- There are several review articles about the same topic published in the past few years, this work did not raise new angles or perspectives towards this area. The only new angle is the discussion related to LUMO and HOMO in section 3.2. However, it was not well structured and hence not easy to follow.
Therefore, this work is less strong for Molecules in the current form.
Author Response
Reviewer 1 Comments |
|||
Comment number |
Comments |
Justification |
Ref. |
1 |
As suggested by the title and abstract that this manuscript focuses on ionic liquids, the general knowledge about CO2 electroreduction (types of reactors, types of products) should focus more on works using ionic liquids to be more relevant.
|
We have incorporated this comment.
|
|
Typical CO2 reduction cells in liquid-based electrolytes system was added, supported by Figure 3 for more clarification |
· Page 2, line 55,56, -65-68. · Page 3- Figure 3.
|
||
New section (2) is added to introduce ionic liquid including products of CO2ERR in liquid-based electrolytes system |
· Page 7- lines 172-189. · Page 8-Figure 5. · Page 8-lines 198-201.
|
||
2 |
There are several review articles about the same topic published in the past few years, this work did not raise new angles or perspectives towards this area. The only new angle is the discussion related to LUMO and HOMO in section 3.2. However, it was not well structured and hence not easy to follow.
|
We have incorporated this comment by restructuring the sentences and by the addition of more explanations. |
· Page 15, lines 353-355,365, 366. Page 16, lines 370-375,377-384,390,391,395 Page 17, lines 426-428. · Page 20, lines 490,491. |
The corrections are highlighted in the manuscript in attachment

Reviewer 2 Report
- Other CO2 reduction methods should be compared in the introduction section and relevant references should be cited such as J. Mater. Chem. A, 2020,8, 12744-12756.
- Some typing mistakes should be corrected.
Author Response
We have carefully reviewed the comments and have revised the manuscript accordingly. Our responses are given in a point-by-point manner below (tables). Changes to the manuscript is highlighted in the review article (attached separately).
Reviewer 2 Comments |
|||
Comment number |
Comments |
Justification |
Ref. |
1 |
Other CO2 reduction methods should be compared in the introduction section and relevant references should be cited such as J. Mater. Chem. A, 2020,8, 12744-12756.
|
We have incorporated this comment. Several technologies on CO2 reduction were added including thermal reduction. Moreover, the potential and limitation of the process was highlighted |
· Page1: Lines 29-39 |
2 |
Some typing mistakes should be corrected. |
We have thoroughly checked and made corrections on spelling, grammatical errors and formatting of the paper.
|
|

Reviewer 3 Report
The authors presents a review on elucidation of the roles of ionic liquid in CO2 electrochemical reduction to value-added chemicals and fuels. My comments are the following:
- The quality of Fig.3 is too low. The texts in Fig. was not clear.
- As the review focus on the role of ionic liquid. A section which introduces the ionic liquids (definition, properties , exemples- cations and anions ) would be necessary.
- Why the ionic liquids are suitable for CO2 conversion?
- Please make a table to summary all the ILs mentioned in this review.
Author Response
We have carefully reviewed the comments and have revised the manuscript accordingly. Our responses are given in a point-by-point manner below (tables). Changes to the manuscript is highlighted in the review article (attached separately).
Reviewer 3 Comments |
|||
Comment number |
Comments |
Justification |
Ref. |
1 |
The quality of Fig.3 is too low. The texts in Fig. was not clear.
|
We have incorporated this comment. Because of the addition of one more figure, the figure number changed to Fig 4. now
|
· Page 5- Fig 4. |
2 |
As the review focus on the role of ionic liquid. A section which introduces the ionic liquids (definition, properties, examples- cations and anions) would be necessary.
|
We have incorporated this comment. New section was added. Section (2)
|
· Page 7-lines 171-174. · Page 7-section (2) Lines 177-188. · Page 8 -Figure 5. · Page 8 - Lines 198-201.
|
3 |
Why the ionic liquids are suitable for CO2 conversion |
We have incorporated this comment and answered the question. |
· Page 7 line 171 and Page 20 Lines 506-512. |
4 |
Please make a table to summary all the ILs mentioned in this review.
|
We have incorporated this comment. We merged the summary of the ILs mentioned in this review with Table 2 (highlighted in green) |
· Page 11- lines 269-271. · Page 11- Table 2- (highlighted in green)
|

Round 2
Reviewer 1 Report
The authors have addressed all the comments, so the manuscript in the present form can be published.